# Development of Chitosan Particles Loaded with siRNA for Cystatin C to Control Intracellular Drug-Resistant *Mycobacterium tuberculosis*

**DOI:** 10.3390/antibiotics12040729

**Published:** 2023-04-08

**Authors:** David Pires, Manoj Mandal, Ana I. Matos, Carina Peres, Maria João Catalão, José Miguel Azevedo-Pereira, Ronit Satchi-Fainaro, Helena F. Florindo, Elsa Anes

**Affiliations:** 1Host-Pathogen Interactions Unit, Research Institute for Medicines, iMed.ULisboa, Faculty of Pharmacy, Universidade de Lisboa, Av. Prof. Gama Pinto, 1649-003 Lisboa, Portugal; 2Center for Interdisciplinary Research in Health, Católica Medical School, Universidade Católica Portuguesa, Estrada Octávio Pato, 2635-631 Rio de Mouro, Portugal; 3Drug Delivery and Immunoengineering Unit, Research Institute for Medicines, iMed.ULisboa, Faculty of Pharmacy, Universidade de Lisboa, Av. Prof. Gama Pinto, 1649-003 Lisboa, Portugal; 4Department of Physiology and Pharmacology, Sackler Faculty of Medicine, Tel Aviv University, Tel Aviv P.O. Box 39040, Israel

**Keywords:** tuberculosis, antibiotic resistance, host-directed therapies, nanomedicines, chitosan, cathepsin inhibitors, cystatins

## Abstract

The golden age of antibiotics for tuberculosis (TB) is marked by its success in the 1950s of the last century. However, TB is not under control, and the rise in antibiotic resistance worldwide is a major threat to global health care. Understanding the complex interactions between TB bacilli and their host can inform the rational design of better TB therapeutics, including vaccines, new antibiotics, and host-directed therapies. We recently demonstrated that the modulation of cystatin C in human macrophages via RNA silencing improved the anti-mycobacterial immune responses to *Mycobacterium tuberculosis* infection. Available in vitro transfection methods are not suitable for the clinical translation of host-cell RNA silencing. To overcome this limitation, we developed different RNA delivery systems (DSs) that target human macrophages. Human peripheral blood-derived macrophages and THP1 cells are difficult to transfect using available methods. In this work, a new potential nanomedicine based on chitosan (CS-DS) was efficiently developed to carry a siRNA-targeting cystatin C to the infected macrophage models. Consequently, an effective impact on the intracellular survival/replication of TB bacilli, including drug-resistant clinical strains, was observed. Altogether, these results suggest the potential use of CS-DS in adjunctive therapy for TB in combination or not with antibiotics.

## 1. Introduction

Human tuberculosis (TB) is a respiratory infectious disease mainly caused by the species *Mycobacterium tuberculosis* (Mtb). Lung disease is the major manifestation of TB compared with less common extrapulmonary forms and is the absolute requirement for effective pathogen transmission via aerosols to new susceptible hosts [1]. In the low respiratory tract, the pathogen engages the innate immune responses. The encounter with macrophages and dendritic cells allows the initial expansion of the pathogen in intracellular niches and the activation of adaptive immunity [2,3,4]. The efficient containment of the infection in the lung is accomplished by cells of the adaptive immune system that will lead to the formation of a solid granuloma and a state of latent infection characterized by no symptoms of disease and no transmission [5]. Active TB occurs either after primary infection or after reactivation from latency, particularly due to malnutrition, diabetes, aging, or immunosuppressive conditions (e.g., HIV co-infection or immunomodulatory therapies) [5]. In this situation, the granuloma progresses to a necrotizing state with uncontrolled pathogen replication, and lung cavitation, all typical hallmarks of the active disease TB [6]. Less often, the granuloma evolves to an impaired structure where the extensive apoptosis of adaptive immune cells leads to extrapulmonary TB [5].

Despite the availability of antibiotics and BCG vaccines, TB remains a leading representative among respiratory tract diseases threatening public health. In the 21st century, TB is still causing more than 1.4 million deaths and 10 million new cases annually [7], only surpassed by the COVID-19 pandemic, which caused 1.8 million deaths in 2020 (WHO Coronavirus Disease (COVID-19) Dashboard. https://COVID19.who.int/, accessed on 12 January 2021). People suffering from this disease are estimated to be a small proportion of all latently infected individuals. Mtb has infected about one-quarter of the world population, with some 600,000 individuals carrying multi- and extensively drug-resistant (MDR and XDR, respectively) strains. Altogether, this is creating a remarkable resource for future cases of TB that will suffer from insufficient treatment options. According to the global TB report from the WHO [7], the treatment success rate of MDR-TB is 56.0 %, while extensively severe drug-resistant TB (XDR-TB) is almost incurable.

Understanding the complex interactions between the pathogen and its host can inform the rational design of less toxic, shortened, and more efficacious TB therapeutics, including new antibiotics and host-directed/adjunctive therapies. The hallmark of latent or active infection, the granuloma, is a challenging dynamic structure [1]. It is formed by several layers of cells with macrophages taking place at the center. Here, Mtb’s ability to survive intracellularly results in cycles of Mtb infection that intercept freshly arrived macrophages, perpetuating the infection cycle [8,9]. Targeting these intracellular niches is central to the control of the infection/colonization. Our group found that one mechanism through which Mtb establishes intracellular niches is by manipulating lysosomal cathepsins and their natural inhibitors cystatins [10,11]. Recently, we overcame the pathogen-induced blockade of cathepsins by using different strategies, including targeting microRNAs [12,13,14], using protease inhibitors such as saquinavir [15,16,17], or targeting cystatin C [18]. Particularly, the RNA silencing of cystatin C culminated with a positive impact on cathepsins’ proteolytic activity, resulting in better anti-mycobacterial immune responses to Mtb infection [18]. This opens up new doors for the delivery of this siRNA as a clinical approach.

Hence, in this work, we developed a delivery system (DS) for cystatin C siRNA, as the usual transfection methods are not suitable for clinical translation. The biomedical application of nanotechnology has opened up the way to combine drugs and corresponding carriers into a new class of pharmaceuticals with a particle size ranging from 1 to 1000 nm [19]. This approach is expected to enhance drug targeting and reduce toxicity while increasing drug stability and improving drug absorption and efficacy [20,21,22,23].

Chitosan (Cs) is a natural non-toxic polymer with demonstrated properties such as biocompatibility and biodegradability, revealing good adhesion to mucosae and direct antibacterial properties for a variety of Gram-positive and Gram-negative species [24,25,26,27,28], which makes it a suitable material for a DS. Here, we provide evidence substantiating a future clinical application of Cs-DS to treat drug-resistant TB by acting as a host-directed adjunctive therapy that improves the intracellular killing of Mtb.

## 2. Results

### 2.1. Physicochemical Characterization of siRNA-Incorporating Particle Formulations

Polymeric DSs were synthesized to deliver siRNA-targeting cystatin C (CstC). First, GlutCs/pARG–siRNA polyplexes were developed to allow the entrapment and subsequent release of the siRNA. The interaction between the positively charged chitosan [29,30] or arginine [31,32] and the negatively charged siRNA leads to the spontaneous formation of stable polyplexes in the aqueous milieu. Accordingly, GlutCs and pARG were used at two different ratios, 15:1 and 2.5:1 (*m*/*m*), for GlutCs/pARG:siRNA, respectively, to prepare the polyplexes through electrostatic-based interactions. To potentiate the interaction of DSs with macrophages, mannose-functionalized DSs were developed to improve their recognition by the mannose receptor (CD206) expressed at the macrophage surface by promoting receptor–ligand interaction and subsequently improving payload delivery [33]. siRNA-loaded DSs presented an average hydrodynamic diameter close to 200 nm, with a low polydispersity index, a near-neutral surface charge, and a spherical shape (Table 1 and Appendix A). Macrophage-targeted NPs displayed high levels of entrapment efficiency and loading capacity for both siRNAs (Table 1).

### 2.2. Particles Loaded with Anti-Cystatin C siRNA Have no Cytotoxic Effects on Human Primary Macrophages

To define the optimal conditions for macrophage treatment with DSs, first, we assessed the highest concentration of DSs without cytotoxic effects using resazurin, and second, we evaluated the long-lasting incubation time with that concentration with no cell viability alterations. Figure 1 shows the results obtained in human monocyte-derived macrophages (HMDMs) treated for 72 h with different concentrations of DSs (Figure 1a) and for different periods of incubation using DSs (Figure 1b) with a fixed concentration of 1 mg/mL. For both formulations of DSs, the highest concentration with no significant cytotoxic effects corresponds to 1 mg/mL. In addition, the exposure to DSs for as long as 72 h did not show any effects on cell viability. A common protocol using a commercial transfection reagent served as a control reference and revealed that DSs produce comparable results to the transfection reagent for 6 h and 24 h, while for 72 h, only the treatment with the transfection reagent produced statistically significant cytotoxic effects.

### 2.3. Chitosan Particles Loaded with Anti-Cystatin C siRNA Effectively Induce Silencing in Primary Human Macrophages and in THP-1 Cells

Having defined the conditions without statistically significant cytotoxic effects, next, the efficacy of these DSs in delivering CstC siRNA and silencing CstC mRNA was evaluated. HMDMs were treated with 1 mg/mL of DSs for distinct periods of time, after which the cells were washed (to renew the culture medium and remove eventual remaining extracellular DSs). Incubation further proceeded until 72 h post-exposure to DSs. The quantification of CstC mRNA using qPCR indicated that Cs-DS-silencing effects were obtained 6 h after exposure, while maximum silencing was achieved by 72 h (Figure 2a). By contrast, for pARG-DS, no silencing was detected, and the treatment produced more erratic effects that, in some cases, would even induce CstC expression rather than silencing.

Since longer periods of exposure produced increased silencing of CstC, we further tested extending the time of contact of the cells to DSs (without the washing step). The results show that Cs-DS effectively silence CstC mRNA (Figure 2b) for 72 h with no benefit in increasing exposure time. Again, the results for pARG-DS were less consistent, and no statistically significant silencing of CstC mRNA was detected. As before, a commercial transfection reagent was used as a control reference, showing the effective silencing of CstC mRNA but with a comparatively less extent of silencing than with Cs-DS.

Besides HMDMs, it was decided to test DSs in macrophages derived from the monocytic cell line THP-1 using the most optimal previously defined treatment conditions of 1 mg/mL for 72 h. This cell line is usually regarded as difficult to transfect. Our results were similar to those achieved for HMDMs with Cs-DS or the transfection reagent, leading to an effective reduction in CstC mRNA levels, while pARG-DS had no effect.

### 2.4. Chitosan Particles Are Efficiently Internalized by Primary Human Macrophages and THP-1 Cells

To control the distribution of DSs in the population of macrophages, we proceeded to analyze the capacity of HMDMs and THP-1 cells to internalize DSs. For this, we labeled Cs-DS with the fluorophore Cy5 to track them via fluorescence. Macrophages were treated with 1 mg/mL of Cy5-labeled Cs-DS for 6 h and then analyzed via flow cytometry and confocal microscopy. Figure 3 shows that for both types of macrophages, almost all cells were positive for DSs, and the results were very consistent, resulting in imperceptible values for standard error (Figure 3c). Concerning the number of DSs internalized, the mean fluorescence levels were one log higher for HMDMs than for THP-1 cells (Figure 3c), indicating an increased capacity of Cs-DS to target these cells. Furthermore, no differences were detectable when cells were treated with empty DSs or with DSs carrying siRNA.

### 2.5. Chitosan Particles Loaded with Anti-Cystatin C siRNA Impact the Intracellular Killing of Mtb Strains Either with Susceptibility or with Distinct Drug-Resistance Profiles

As mentioned, we have previously shown that the targeting of CstC by siRNA improves the macrophages’ response to Mtb infection [18]. Likewise, here, we proceeded to evaluate the effect using Cs-DS as a delivery system of siRNA. For this, HMDMs were treated with 1 mg/mL of Cs-DS carrying CstC siRNA for 72 h prior to infection with Mtb. In these experiments, we compared different strains of Mtb, including the reference laboratory strain H37Rv and three clinical strains: one susceptible to drug therapy, one multi-drug-resistant (MDR), and one extensively drug-resistant (XDR) strain. The characterization of these strains was performed by the National Institute of Health (INSA, Portugal). This enabled us to account for strain variability and the potential outcomes related to mutations conferring drug resistance. Intracellular Mtb was quantified by measuring the colony-forming units recovered from infected macrophages over a 7-day period. The results show that for all the strains tested, treatment with Cs-DS loaded with CstC siRNA results in a significant impact on the intracellular burden of Mtb compared with Cs-DS carrying the scramble siRNA control (Figure 4a). This reduction in the intracellular survival of Mtb was statistically significant from day 3 of infection onwards and had further impacts over time. Furthermore, it was found that even though the levels of CstC mRNA decreased over time after infection in the scramble control (as expected), Cs-NP CstC siRNA treatment was able to further impact the extent of the decrease in CstC mRNA for 72 h after all DSs were extracellularly washed out (Figure 4b). Moreover, to verify that mRNA silencing translates to lower CstC protein levels, we quantified the levels of CstC protein at 72 h post-treatment with DSs. The results in Figure 4b, right panel, indicate a 60% reduction in CstC protein expression, confirming the efficacy of NP-induced silencing.

## 3. Discussion

Cystatin C is a type-II cystatin and acts as an endogenous regulator of proteolytic activity by competitively and reversibly binding to the active site of several cathepsins, preventing substrate binding [34]. CstC is one of the most potent inhibitors of cysteine proteases such as human cathepsins B, H, L, and S, having inhibition constants below the nanomolar range [34]. It is widely expressed in human cells and tissues and abundantly secreted, being found in high levels in body fluids and serving as a biomarker for several diseases and body functions [35,36]. In innate immune phagocytes, this cystatin accumulates in endosomal vesicles, where it colocalizes with active cathepsins [37]. CstC was shown to significantly increase Mtb intracellular survival in primary human macrophages, in a similar way to general cysteine cathepsin inhibitor E-64d [10]. This impact on Mtb survival is most likely due to the inhibition of cathepsin activity since the shRNA screening of most cathepsins revealed that more than half of these proteases participate in Mtb killing [10]. This is added to the fact that Mtb infection has been shown to induce a broad downregulation of cathepsin activity that is not replicated by non-pathogenic mycobacteria and is hypothesized to be a mechanism for enhanced Mtb intracellular survival. The strong inhibitory effect of CstC on cysteine cathepsins, and its elevated basal levels of expression, make it a promising target for the manipulation of the host-cell proteolytic activity. In fact, we have recently demonstrated the benefits of siRNA silencing of CstC in primary human cells infected with Mtb and co-infected with HIV [18]. By silencing CstC mRNA, we were able to restore the proteolytic activity in infected macrophages with consequences in Mtb killing and antigen presentation to T lymphocytes.

RNA-targeting therapeutics have the potential to modulate the relevant genes that otherwise would not be targetable by small molecules and proteins. Interestingly, the first RNA-targeting antisense oligo (ASO), approved for medication in 1998, was used to treat an infectious disease, namely, cytomegalovirus retinitis in AIDS patients [38]. To our knowledge, and with the exception of mRNA vaccines, there are no other RNA-based therapies approved against infectious diseases. Concerning RNA interference (RNAi) therapies, as the one here proposed, 2018 marked the first approval of an RNAi drug in the USA, patisiran, for the treatment of the polyneuropathy of hereditary TTR-mediated amyloidosis, and since then, several drugs have been approved in the USA and Europe and many more are in late phase 3 clinical trials [39,40].

A challenge faced by these types of drugs is the unstable nature of RNA in the serum and intracellularly, as well as its poor cell penetration [33]. The need to protect RNA from these harmful physiological environments stimulated the development of nanoparticle drug delivery systems, particularly lipid nanoparticles and cationic polymer-based polyplexes. Among the latter, chitosan-based nanoparticles have been used for RNA and siRNA delivery and have demonstrated several beneficial properties such as low toxicity, biocompatibility, biodegradability, and permeability [30,39]. Likewise, polyarginine DSs have been developed for siRNA delivery with low toxicity [31,32]. The results here pointing to the use of Cs-DS and pARG-DS in primary human macrophages support this evidence of low toxicity since exposure to Cs-DS in concentrations as high as 1 mg/mL for at least 72 h did not produce detectable cytotoxic effects.

There have been reports of limited transfection efficiency of some nanoparticle systems, particularly with chitosan [26,30]. One solution to overcome this limitation is the functionalization of nanoparticles with ligands for highly expressed receptors on the target cell. The DSs used in the present study were functionalized with mannose to potentiate the interaction with the mannose receptor (CD206) expressed on macrophages by promoting receptor–ligand interaction and subsequently improving siRNA delivery [29,33]. Our previous study indicated the ability of these mannosylated particles in following the endolysosomal pathway, in addition to being able to overcome endosomal degradation and thereby release payloads into the cytoplasm [41]. Therefore, these particles were the starting point of this study focused on the regulation of CstC function and its impact on macrophages’ response to Mtb infection.

For Cs-DS, the results for CstC silencing efficacy, from flow cytometry and confocal microscopy, demonstrate the success of this approach in targeting primary macrophages and in macrophages derived from the monocytic cell line THP-1, both regarded as difficult to transfect. Cs-DS carrying CstC siRNA could effectively silence CstC mRNA after 6 h of treatment and for at least 96 h. By contrast, pARG-DS did not produce significant silencing of CstC, and their inconsistent results led to the selection of Cs-DS for further testing.

Cs-DS was tested in the context of Mtb infection by the reference strain H37Rv and by three clinical strains with different phenotypes of drug resistance. The results show an effective capacity to silence CstC for at least 72 h during infection and following six days from the single-dose administration of Cs-DS. This treatment resulted in improved intracellular killing for all strains, despite their different drug-resistance profiles and growth kinetics. This result is in line with our previous evidence using common commercial transfection reagents [13,18]. In fact, taking into consideration (1) that common in vitro cell transfection methods are not translatable for the clinic; (2) that the present DS was demonstrated to be more efficient in silencing cystatin C mRNA over 96 h; and finally, (3) that it targets the siRNA to macrophages via mannose receptor where it overcomes endolysosomal degradation, it is evident they have the potential for application in complex host organisms.

Altogether, these results demonstrate the potential for a chitosan-particle-based solution to modulate cystatin C expression, overcome the Mtb-induced blockade of the proteolytic function of macrophages, and improve the control of the infection. Further studies will be necessary to determine if the results obtained in this study can be replicated in vivo and to evaluate the pharmacokinetics of the treatment. It would also be of significant interest to the TB field to evaluate the efficacy of this treatment during latent Mtb infection. Here, we propose this delivery system for silencing cystatin C as a potential host-directed therapy that can be applied to complement the current antibiotic therapy and overall contribute to overcoming drug resistance.

## 4. Materials and Methods

### 4.1. Preparation and Physicochemical Characterization of the Particle Formulations

#### 4.1.1. Materials and Reagents

Poly(L-lactic acid) (PLA) (2,000 Da) with a weight-averaged molecular mass (Mw) of 2000 was purchased from PolySciences Europe GmbH. Poly(lactic-co-glycolic acid (PLGA)-mannose (PLGA-man) was synthesized and characterized based on Conniot et al. [41]. PLGA (Resomer 503H, Mw 24,000–38,000), *D*-mannosamine hydrochloride (mannosamine·HCl), dimethylformamide, 4-dimethylaminopyridine, *N,N′*-dicyclohexylcarbodiimide, methanol, anhydrous sodium sulfate, poly(vinyl alcohol) (PVA, M_w_ 13,000–23,000 Da, 99% hydrolyzed), D-alpha-tocopherol polyethylene glycol 1000 succinate (TPGS), Pluronic^®^ F-127 (PF-127), dichloromethane (DCM), and deuterated chloroform (CDCl_3_) were purchased from Sigma-Aldrich. *N*-butyl poly-L-arginine hydrochloride (pARG, M_w_ range 3000–3400) was purchased from Polypeptide Therapeutic Solutions (Valencia, Spain). Glutamate chitosan (GlutCs; Protasan UP G113) was purchased from NovaMatrix (Sandvika, Norway). Quant-iT™ RNA Assay Kit was purchased from Thermo Fisher Scientific (Waltham, MA, USA). Agarose, tris-acetate-EDTA (TAE) 50× buffer, and the loading buffer were purchased from VWR Scientific (Radnor, PA, USA).

#### 4.1.2. Preparation of Chitosan/Arginine–siRNA Polyplexes

GlutCs, pARG, and siRNA were dissolved in RNase-free water at 20.13, 3.355, and 1.342 mg mL^−1^ (100 mM), respectively. GlutCs/pARG–siRNA polyplexes were formed by quickly mixing the siRNA solution with an equal volume of GlutCs (15:1 (*m*/*m*) GluCs:siRNA ratio) or pARG (2.5:1 (*m*/*m*) pARG:siRNA ratio) solution dropwise. This mixture was further incubated under slow stirring, for 1 h at room temperature.

#### 4.1.3. Synthesis of Polymeric Multifunctional DSs

PLGA-man/PLA DSs were prepared via a double-emulsion (water-in-oil-in-water (*w/o/w*)) solvent-evaporation method [41]. A PLGA-man/PLA (2:8) blend was dissolved in DCM at 50 mg mL^−1^. A 10% (*m*/*v*) PVA aqueous solution that contained 100 µM of siRNA previously complexed with GlutCs or pARG (100 µL) was added to the organic phase containing the polymer blends dissolved in DCM. The internal aqueous phase used for the synthesis of empty NPs contained the GlutCs or pARG dissolved in the 10% (*m*/*v*) PVA. The mixture was emulsified under continuous sonication at 20% of amplitude for 15 s, using a microprobe ultrasonic processor. A second emulsion was performed by adding a 2.5% (*m*/*v*) TPGS aqueous solution (400 µL) to that w/o emulsion under the same conditions. The resultant w/o/w double emulsion was subsequently added dropwise into a 0.125% (*m*/*v*) PF-127 aqueous solution and stirred for 1 h at room temperature. NPs were separated via centrifugation at 20,000× *g* for 45 min at 4 °C (Beckman Coulter Allegra 64R High-Speed Centrifuge, Brea, CA, USA), washed with ultrapure water, and resuspended in PBS. Cy5-labeled NPs were prepared by adding 2.5 mg mL^−1^ of Cy5-grafted PLGA to the polymer blend.

#### 4.1.4. Size Distribution and ζ Potential Measurements

A Zetasizer Nano ZS instrument (Malvern Instruments, Malvern, UK) was used to determine the NP hydrodynamic mean diameter and polydispersity index (PdI) via dynamic light scattering [41]. The same equipment allowed for the determination of ζ potential of NPs measured via laser Doppler velocimetry in combination with phase analysis light scattering [41]. NPs were diluted in PBS, and their Brownian motion based on laser light scattering (NP size) and electrophoretic mobility using the Helmholtz–von Smoluchowski model (ζ potential) were determined at 25 °C via cumulative analysis [41].

#### 4.1.5. Entrapment Efficiency and Loading Capacity of siRNA

The amount of siRNA anti-cystatin C (CstC siRNA) and negative control (scramble) entrapped in NPs was indirectly quantified in the supernatants collected from centrifugation following NP preparation. Entrapment efficiency (EE, Equation (1)) and loading capacity (LC, Equation (2)) were quantified using a Quant-iT™ RNA Assay Kit (broad range), following the manufacturer’s instructions. The relative fluorescence for the RNA assay kit was measured using a microplate reader (FLUOstar Omega, BMG Labtech, Ortenberg, Germany) at 644 nm excitation and 673 nm emission wavelengths.
(1)$$\mathrm{EE}\;\left(\%\right)=\frac{\mathrm{initial}\;\mathrm{amount}\;\mathrm{of}\;\mathrm{siRNA}-\mathrm{amount}\;\mathrm{of}\;\mathrm{siRNA}\;\mathrm{in}\;\mathrm{the}\;\mathrm{supernatant}}{\mathrm{initial}\;\mathrm{amount}\;\mathrm{of}\;\mathrm{siRNA}}\times 100$$
(2)$$\mathrm{LC}\;\left({µg\;\mathrm{mg}}^{-1}\right)=\frac{\mathrm{initial}\;\mathrm{amount}\;\mathrm{of}\;\mathrm{siRNA}\;-\;\mathrm{amount}\;\mathrm{of}\;\mathrm{siRNA}\;\mathrm{in}\;\mathrm{the}\;\mathrm{supernatant}}{\mathrm{total}\;\mathrm{amount}\;\mathrm{of}\;\mathrm{polymer}}$$

### 4.2. Cell Isolation and Culture Conditions

Human monocyte-derived macrophages (HMDMs) were isolated and differentiated from buffy coats of healthy human donors provided by the National Blood Institute (Instituto Português do Sangue e da Transplantação, I.P., Lisbon, Portugal) as previously described [13]. The human monocytic cell line THP-1 (ATCC TIB202) (American Type Culture Collection, VA, USA) was cultivated as previously described [42] and was differentiated to macrophages via incubation with 20 nM phorbol 12-myristate 13-acetate (PMA) overnight.

### 4.3. Macrophage Treatment

Macrophages were treated with GlutCs particles (Cs-DS) or pARG particles (pARG-DS) loaded with anti-cystatin C (CstC) siRNA SMARTpool ON-TARGETplus human CST3 siRNA (Agilent Technologies, Inc, Santa Clara, CA, USA) with target sequences (from 5′ to 3′) CAAUGACCUUGUCGAAAUC, CGUCGGCGAGUACAACAAA, GAACCACGUGUACCAAGAC, and UAGCUAGGGGUGAACUACUU or the respective siRNA non-targeting (scramble) control (Agilent Technologies, Inc, Santa Clara, CA, USA), with target sequences (from 5′ to 3′) UGGUUUACAUGUCGACUAA, UGGUUUACAUGUUGUGUGA, UGGUUUACAUGUUUUCUGA, and UGGUUUACAUGUUUUCCUA. For comparison, macrophages were transfected with the same siRNAs using a ScreenFectA (ScreenFect GmbH, Eggenstein-Leopoldshafen, Germany) transfection reagent and following the manufacturer’s protocol. Macrophages were incubated with the transfection reagent and 100 nM of siRNA, which was calculated to be the same concentration of siRNA present in 1 mg/mL of DS.

### 4.4. Macrophage Viability

Macrophages treated with DSs were incubated with a 10% (*v*/*v*) of PrestoBlue (Invitrogen, Carlsbad, CA, USA) resazurin-based solution at 37 °C and 5% CO_2_, for 3 h. Fluorescence was quantified according to the manufacturer’s instructions using a Tecan M200 Pro spectrofluorometer (Tecan, Männedorf, Switzerland). Non-treated macrophages served as references for 100% viability, and Igepal-treated macrophages (0.05%) were used as references for 0% viability.

### 4.5. qPCR

RNA was isolated using an NZY Total RNA Isolation Kit (NZYTech, Lisbon, Portugal), following the manufacturer’s instructions. The reverse-transcriptase reaction was performed from 100 ng of total RNA using an NZY First-Strand cDNA Synthesis Kit (NZYTech, Lisbon, Portugal), according to the manufacturer’s instructions. qPCR was performed using an NZY qPCR Green Master Mix (NZYTech, Lisbon, Portugal) with a primer set specific for CST3 mRNA (F-CAACAAAGCCAGCAACGACAT; R-AGAGCAGAATGCTTTCCTTTTCAGA) and for glyceraldehyde 3-phosphate dehydrogenase (GAPDH) mRNA (F-AAGGTGAAGGTCGGAGTCAA; R-AATGAAGGGGTCATTGATGG), according to previously described conditions [18]. The qPCR was performed using a QuantStudio™ 7 Flex System (Thermo Fisher Scientific, Waltham, MA, USA). Data were analyzed using the ΔΔCt method. The mRNA expression profiles were normalized to the GAPDH housekeeping gene and finally calculated relative to the scramble control siRNA-treated samples.

### 4.6. Flow Cytometry

Macrophages treated with cyanine-5-labeled DSs were detached using Accutase and fixed in 4 % paraformaldehyde for 15 min before being analyzed using an Accuri C6 Plus flow cytometer (BD, Franklin Lakes, NJ, USA). Non-labeled DSs were used for reference. Data analysis was performed in FCS Express 7 (De Novo Software, Pasadena, CA, USA).

### 4.7. Confocal Microscopy

Macrophages treated with cyanine-5-labeled DSs were fixed with 4% paraformaldehyde for 15 min and quenched with 50 mM of NH_4_Cl in PBS for 15 min. Cells were permeabilized with 0.1% Triton X-100 for 5 min and counter-stained with DAPI (Thermo Fisher Scientific, Waltham, MA, USA). Coverslips were mounted using ProLong Diamond Antifade Mountant (Thermo Fisher Scientific, Waltham, MA, USA) and visualized on a Leica TCS SP8 confocal microscope (Leica Camera AG, Wetzlar, Germany). Analysis was performed using the Leica Application Suite X (Leica Camera AG, Wetzlar, Germany) and Fiji software [43].

### 4.8. Bacterial Cultures and Infection Procedure

*M. tuberculosis* H37Rv (ATCC 27294) (American Type Culture Collection, Manassas, VA, USA) and three clinical strains provided and characterized by the Portuguese National Institute of Health (Instituto Nacional de Saúde Doutor Ricardo Jorge (INSA)) were cultivated in Middlebrook’s 7H9 medium supplemented with a 10% oleic acid albumin dextrose (OADC) supplement (BD Difco, Franklin Lakes, NJ, USA), 0.02% glycerol, and 0.05% tyloxapol (Merck, KGaA, Darmstadt, Germany) at 37 °C. The clinical strains were isolated from patients with active TB. The drug-susceptible clinical strain (INSA code 33427) is susceptible to streptomycin, isoniazid, rifampicin, and pyrazinamide; the MDR strain (INSA code 34192) is resistant to the previous antibiotics plus ethionamide; the XDR strain (INSA code 163761) is resistant to all the previous antibiotics plus amikacin, kanamycin, capreomycin, moxifloxacin, and ofloxacin. All Mtb strains were manipulated in the biosafety level 3 laboratory at the Faculty of Pharmacy of the University of Lisbon (Lisbon, Portugal), following the respective national and European biosecurity standards, based on applicable EU Directives.

### 4.9. Infection Procedure

Bacteria were cultivated for approximately 7 days at 37 °C until reaching the exponential growth phase. The bacterial suspensions were centrifuged and washed in phosphate-buffered saline (PBS) and resuspended in a macrophage culture medium without antibiotics. Bacterial clumps in suspension were dismantled using an ultrasonic bath for 5 min and removed via low-speed centrifugation at 500× *g* for 1 min. The final single-cell suspension was verified through microscopy and quantified by measuring the optical density at 600 nm. The macrophages were infected with a multiplicity of one bacterium per macrophage for 3 h at 37 °C and 5% CO_2_. Following this period, free bacteria were removed by washing the macrophages with PBS and by adding a fresh complete medium.

### 4.10. Bacterial Intracellular Survival

Intracellular bacteria were recovered by lysing the infected macrophages, at the selected time points of infection, using a 0.05% Igepal solution (Merck, KGaA, Darmstadt, Germany) for 15 min. The serial dilutions of the resulting bacterial suspension were plated in Middlebrook’s 7H10 solid medium, supplemented with 10% OADC (BD Difco, Franklin Lakes, NJ, USA), and incubated for 2–3 weeks at 37 °C until colonies could be observed and counted under a microscope.

### 4.11. Western Blotting

Total proteins were harvested using a RIPA buffer (Merck, KGaA, Darmstadt, Germany). Prior to electrophoresis, the samples were diluted 1:1 in a Laemmli buffer (Merck, KGaA, Darmstadt, Germany) and heated at 95 °C for 5 min. Proteins were separated using sodium dodecyl sulfate–polyacrylamide gel electrophoresis (SDS–PAGE) using TGX FastCast 10% acrylamide gels (Bio-Rad Laboratories, Hercules, CA, USA) and transferred to the nitrocellulose membrane using the Trans-Blot Turbo Transfer System (Bio-Rad Laboratories, Hercules, CA, USA). The membrane was processed and stained using the iBind Western system (Thermo Fisher Scientific, Waltham, MA, USA), and the antibodies specific for CstC (1:1000 dilution of #ab133495, Abcam, Cambridge, UK), β-tubulin (1:2000 dilution of #ab6046, Abcam, Cambridge, UK), and horseradish peroxidase (HRP)-conjugated secondary antibody (1:2000 dilution of #1706515, Bio-Rad Laboratories, Hercules, CA, USA). The bands were visualized via chemiluminescence using an NZY Supreme ECL HRP substrate (NZYTech, Lisbon, Portugal) in an iBright CL1500 Imaging System (Thermo Fisher Scientific, Waltham, MA, USA). The quantification of band intensity was performed in Fiji software.

### 4.12. Statistical Analysis

Statistical analysis was conducted in GraphPad Prism 9 (GraphPad Software, San Diego, CA, USA). Multiple group comparisons were performed using one-way ANOVA followed by a Holm–Sidak post hoc test. Two group comparisons were performed using Student’s *t*-test. Differences were considered significant when the calculated adjusted-*p* value was equal to or below the alpha level of 0.05.

## Figures and Tables

**Figure 1 antibiotics-12-00729-f001:**
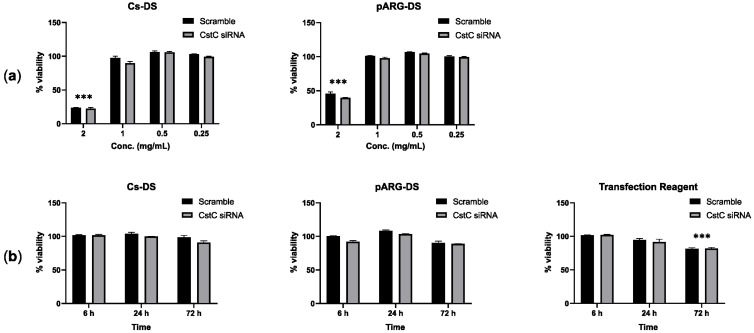
Effect of Cs and pARG particles on the viability of human monocyte-derived macrophages: (**a**) Macrophages were treated with different concentrations of DSs for 72 h or (**b**) treated with 1 mg/mL of DSs for different periods. The tested DSs were loaded with anti-CstC siRNA or scramble control siRNA. A transfection reagent (ScreenfectA) was used for comparison. Macrophage viability was measured using PrestoBlue (resazurin-based solution) by quantifying the fluorescence emission in a plate reader. Results were calculated relative to untreated macrophages (100% viability) and 0.05 % Igepal-treated macrophages (0% viability). Bars represent the average of three independent experiments, and the error bars depict the standard error of the mean. *** *p* ≤ 0.001 is relative to the control and all other concentrations tested (**a**) or to all other treatment times (**b**).

**Figure 2 antibiotics-12-00729-f002:**
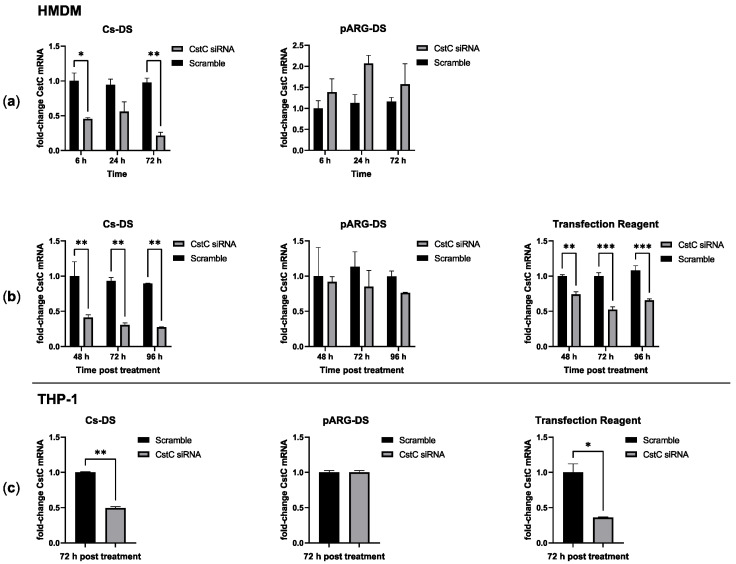
Cystatin C mRNA-silencing efficacy of Cs and pARG particles loaded with anti-CstC siRNA: (**a**) CstC mRNA levels were measured using qPCR in HMDMs treated with 1 mg/mL of DSs for different periods, after which the cells were washed and incubated until 72 h post-treatment; (**b**) alternatively, cells were left in contact with DSs for the selected periods; (**c**) THP-1 macrophages were also treated under the same conditions but only for 72 h. A transfection reagent (ScreenfectA) was used for comparison. Bars represent the average of three independent experiments, and the error bars depict the standard error of the mean. For each plot, the values are presented relative to the scramble control from the earliest time point. * *p* ≤ 0.05, ** *p* ≤ 0.01, and *** *p* ≤ 0.001.

**Figure 3 antibiotics-12-00729-f003:**
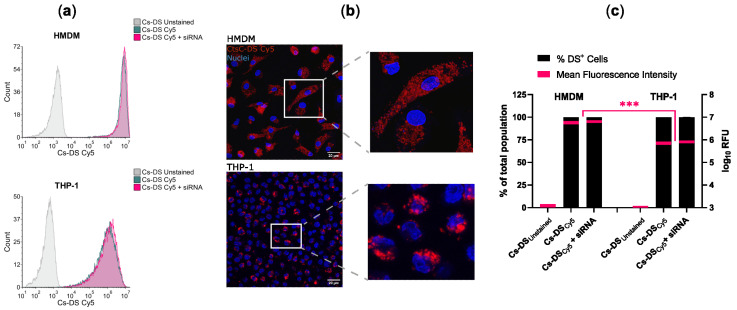
Cs-DS is effectively internalized by human monocyte-derived macrophages and THP-1 macrophages. Macrophages were treated for 6 h with Cy5-labeled empty Cs-DS or siRNA-loaded DSs, and their NP content was analyzed using (**a**) flow cytometry and (**b**) confocal microscopy: (**a**) flow cytometry histograms were obtained from one representative experiment; (**b**) confocal images from one representative experiment. DSs labeled with Cy5 are shown in red, and the cell nuclei labeled with DAPI are in blue; (**c**) the bar plot represents the percentage of macrophages loaded with DSs (left y-axis), and the red lines within the bars represent the mean fluorescence intensity of DSs per macrophage (right y-axis) calculated from three independent samples via flow cytometry. The thickness of the red lines represents min/max data dispersion. *** *p* ≤ 0.001.

**Figure 4 antibiotics-12-00729-f004:**
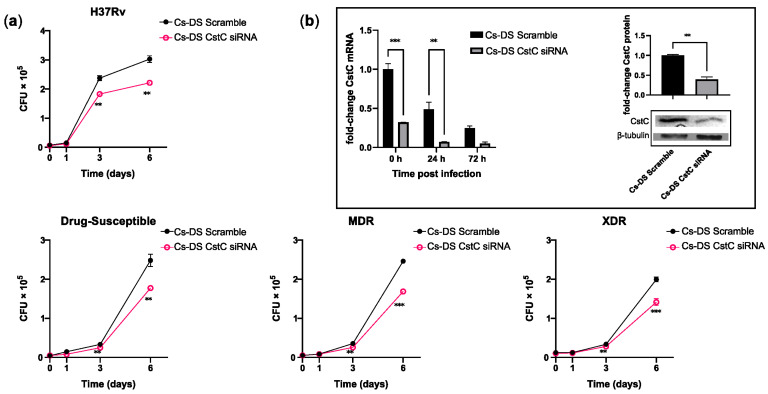
Cs-DS loaded with anti-cystatin C siRNA improves the intracellular killing of laboratory and clinical strains of Mtb with different drug-resistance phenotypes in human macrophages. Human monocyte-derived macrophages were treated with Cs-DS loaded with anti-CstC siRNA or scramble control siRNA 72 h before infection with (**a**) the laboratory strain H37Rv and three clinical strains of Mtb with different levels of drug resistance. Bacterial intracellular survival was evaluated at discrete time points. Line plots depict the average CFU per sample; (**b**) the bar plot of CstC mRNA levels demonstrates the silencing efficacy of Cs-DS during infection with H37Rv. Western blot image demonstrates the silencing of CstC protein by Cs-DS at the moment of infection (t = 0 h). The respective bar plot was calculated from band intensity using β-tubulins as a calibrator. The values depicted are the average of three independent experiments. Error bars represent the standard error of the mean. ** *p* ≤ 0.01, and *** *p* ≤ 0.001.

**Table 1 antibiotics-12-00729-t001:** Physicochemical characterization of the particle formulations.

**Formulation**	**Size (nm)**	**Polydispersity Index**	**ζ-Potential (mV)**	**Entrapment Efficiency (%)**	**Loading Capacity (µg mg^−1^)**
CstC siRNA	194 ± 4	0.095 ± 0.008	−1.38 ± 0.42	96.44 ± 3.98	1.30 ± 0.05
Scramble	192 ± 0	0.112 ± 0.015	−1.37 ± 0.33	99.10 ± 0.77	1.33 ± 0.01

## Data Availability

Not applicable.

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
