# Peer review of "Development of Chitosan Particles Loaded with siRNA for Cystatin C to Control Intracellular Drug-Resistant Mycobacterium tuberculosis"

_antibiotics, 2023, doi:10.3390/antibiotics12040729_

Round 1
Reviewer 1 Report
Dear colleagues, find attached my comments which could help to improve your work.
First in your whole study, I think you should speak about particles and not nanoparticles. Most researchers agree that nanoparticles range in size from 0-100 nm. In your study you have particles because the size is greater than 100 nm
About the title (Line 3), kindly write the full name of M. tuberculosis (Mycobacterium tuberculosis) and as I suggested above, replace nanoparticles by particles ….
In your introduction, please add more information (before line 36) about the disease itself before speaking about the issues related to the treatment (some general information you may use can be find here : https://www.cdc.gov/tb/publications/factsheets/general/tb.htm#:~:text=Tuberculosis%20(TB)%20is%20a%20disease,they%20do%20not%20get%20treatment.)
Line 62: Add a reference after “immune responses to Mtb infection.”
From line 70-72: This sentence must be formulated. It looks like the conclusion or the summary. The introduction is not meant to describe the results.
Let move to material and methods before coming back to the results and discussion
Lines 286, 287…….. : Each abbreviation used for the first time in the text must be written in full. Kindly check in the whole document
Lines 300-301: Add mg mL-1 after 20.13 or remove it after 3.355.
Line 302: What is the meaning of m/m in “15:1 (m/m) 302 GluCs:siRNA ratio)” since you mixed two solutions. The same for Parg..
Lines 322-329: Kindly add the specific analysis conditions or add a reference
Line 330: Kindly replace EE and LC by Entrapment efficiency and loading capacity
Lines 393, 394 and 440: Kindly add the information about Leica Application 393 Suite X and Fiji soft-440 ware as you did in line 386 for FCS Express 7.
Line 412: instead of (or in addition to) “exponential growth”, kindly specify the operating culture conditions (time and temperature)
About the results
The word nanoparticles should not be employed in this work referring to the diameters presented in table 1 (line 97)
Lines 109-113: A comparison test should be done to check if the difference in toxicity between the different treatment times is statistically significant.
Line 125: “Without cytotoxic effects” and “with less cytotoxic effects” are two different things. Kindly check and improve accordingly
Line 138: at which P. value?
Line 141: Instead of “we decided”, kindly add “It was decided”
Good luck
Author Response
Dear colleagues, find attached my comments which could help to improve your work.
First in your whole study, I think you should speak about particles and not nanoparticles. Most researchers agree that nanoparticles range in size from 0-100 nm. In your study you have particles because the size is greater than 100 nm
We appreciate the suggestion of the reviewer which is in line with the nanomedicine definition of the European Science Foundation: “In this context, nanoscale includes active components or objects ranging in size from 1 nm to 100s of nm.” We decided to revise the manuscript and use the terms “particles” and drug “delivery systems (DS)” instead of “nanoparticles”, although many papers and reviews use the term nanoparticle (NP) for particles with a mean average size higher than 100 nm.
About the title (Line 3), kindly write the full name of M. tuberculosis (Mycobacterium tuberculosis) and as I suggested above, replace nanoparticles by particles ….
The title was altered as suggested by the reviewer.
In your introduction, please add more information (before line 36) about the disease itself before speaking about the issues related to the treatment (some general information you may use can be find here : https://www.cdc.gov/tb/publications/factsheets/general/tb.htm#:~:text=Tuberculosis%20(TB)%20is%20a%20disease,they%20do%20not%20get%20treatment.)
We thank the reviewer for this pertinent observation. We have added a paragraph to the introduction to present the characteristics of tuberculosis and Mtb infection during latency or active disease.
Line 62: Add a reference after “immune responses to Mtb infection.”
Reference [18] was introduced as requested.
From line 70-72: This sentence must be formulated. It looks like the conclusion or the summary. The introduction is not meant to describe the results.
We removed this sentence as the reviewer suggested and just added a small portion of text to the paragraph describing chitosan to make it clear why we opted for this material.
Let move to material and methods before coming back to the results and discussion
Lines 286, 287…….. : Each abbreviation used for the first time in the text must be written in full. Kindly check in the whole document
We thank the reviewer for noting these. We have introduced the abbreviations in full text as suggested.
Lines 300-301: Add mg mL-1 after 20.13 or remove it after 3.355.
We removed mg mL-1 after 3.355 as suggested.
Line 302: What is the meaning of m/m in “15:1 (m/m) 302 GluCs:siRNA ratio)” since you mixed two solutions. The same for Parg.
We thank the reviewer for this comment. A brief explanation is presented below to clarify this point. The ability of GluCs or pARG to retard the mobility of siRNA was analyzed by electrophoresis on an electrophoretic mobility shift assay (EMSA). Accordingly, the GluCs:siRNA and pARG:siRNA mass ratios (m/m) were determined on the agarose gel by using free siRNA, as a control. While pARG efficiently retained the siRNA with an optimal pARG:siRNA mass ratio of 2.5:1 (m/m), the optimal GluCs:siRNA mass ratio was 15:1 (m/m). Taking into consideration the EMSA results obtained, to form the GluCh:siRNA polyplex (total volume of 20 µl), 13.42 µg of siRNA (10 µl siRNA at 1.342 mg ml-1, RNase-free water) were added to 201.3 µg of GluCh (10 µl GluCh at 20.13 mg ml-1, RNase-free water), to allow the formation of the GluCs:siRNA complex. On the other hand, to form the pARG:siRNA polyplex (total volume of 20 µl), 13.42 µg of siRNA (10 µl siRNA at 1.342 mg ml-1, RNase-free water) were added to 33.55 µg of pARG (10 µl pARG at 3.355 mg ml-1, RNase-free water), to allow the formation of the pARG:siRNA complex.
Lines 322-329: Kindly add the specific analysis conditions or add a reference.
The reference [41] “Conniot, J.; Scomparin, A.; Peres, C.; Yeini, E.; Pozzi, S.; Matos, A.I.; Kleiner, R.; Moura, L.I.F.; Zupančič, E.; Viana, A.S.; et al. Immunization with Mannosylated Nanovaccines and Inhibition of the Immune-Suppressing Microenvironment Sensitizes Melanoma to Immune Checkpoint Modulators. Nat Nanotechnol 2019, 14, 891–901, doi:10.1038/S41565-019-0512-0.” was introduced as requested (lines 356-363).
Line 330: Kindly replace EE and LC by Entrapment efficiency and loading capacity
Altered as suggested.
Lines 393, 394 and 440: Kindly add the information about Leica Application 393 Suite X and Fiji soft-440 ware as you did in line 386 for FCS Express 7.
We have added information on the Leica microscope and software. Regarding Fiji, this is an open-source project maintained by the community. Since it is based on a scientific publication, the respective paper is cited in the text.
Line 412: instead of (or in addition to) “exponential growth”, kindly specify the operating culture conditions (time and temperature)
As requested, we have introduced this information.
About the results
The word nanoparticles should not be employed in this work referring to the diameters presented in table 1 (line 97)
We agree with the reviewer and therefore, as mentioned above, we replaced the word nanoparticles (NP) with particles and delivery systems (DS).
Lines 109-113: A comparison test should be done to check if the difference in toxicity between the different treatment times is statistically significant.
Indeed, statistical tests were performed to test the variables “scramble vs CstC siRNA”, “Concentration”, and “Treatment time” and their interactions. The only positive results were for the “concentration” variable in Fig1a (2mg/mL vs all others) and for the “treatment time” variable (72h vs all others for Transfection Reagent). To make it clearer what is the comparison corresponding to the *** symbols, we added this to the figure legend: “*** p ≤ 0.001 relatively to the control and all other concentrations tested (a) or to all other treatment times (b).”
Line 125: “Without cytotoxic effects” and “with less cytotoxic effects” are two different things. Kindly check and improve accordingly.
To add clarity to the text, we changed the text in section 2.3 to “without statistically significant cytotoxic effects” and we also changed the text in 2.2 to “while for 72 h only the treatment with the transfection reagent produced statistically significant cytotoxic effects.”
Line 138: at which P. value?
The statistical significance threshold is mentioned in the Methods section and corresponded to a P value of 0.05, below which we considered differences significant. For this specific plot, the P value was 0.49 for the treatment factor (Scr vs CstC) and for each comparison within the same treatment time, the P values were always above that value as well.
Line 141: Instead of “we decided”, kindly add “It was decided”
We altered the text as suggested by the reviewer.
Good luck
Thank you for your help in improving our manuscript.
Reviewer 2 Report
Cystatin C is a type II cystatin and acts as an endogenous regulator of proteolytic activity by competitively and reversibly binding to the active site of several cathepsins, preventing substrate binding. In this work, the Cs-NPs were tested in the context of Mtb infection by the reference strain H37Rv and by three clinical strains with different phenotypes of drug resistance. The results show an effective capacity to silence CstC for at least 72 h during infection and following six days from the single-dose administration of Cs-NPs. This treatment resulted in improved intracellular killing for all strains, despite their different drug-resistance profiles and growth kinetics. This result is in line with our previous evidence using common commercial transfection reagents.
In summary, the results demonstrate the potential for a chitosan-nanoparticle-based solution to modulate cystatin C expression, overcome Mtb-induced blockade of the proteolytic function of macrophages and improve the control of the infection. We propose this delivery system for silencing cystatin C as a potential host-directed therapy that can be applied to complement the current antibiotic therapy and overall contribute to overcoming drug resistance. I think this paper is well organized, and the results are interesting and suitable for publication in Antibiotics. Before further consideration, some minor questions should be fully addressed.
1. There are two ratios (15:1 and 2.5:1 (m/m) for GlutCs/pARG:siRNA) were employed in this work for polymeric NPs. Why did authors select these two ratios, will the ratio above 15:1 or below 2.5:1 significantly affect the resultant properties?
2. I check the Reference 12 (published by the same research group), the targeting of CstC by siRNA improves the macrophages’ response to Mtb infection. So, what’s the innovation of this work based ob polymeric NPs? In the discussion, more descriptions should be highlighted.
3. Please check the Section IV (Experimental section), where did the macrophages and monocytes purchase?
4. Some minor grammar issue. Page 8, line 270. Add a “.” in “….drug resistance The results…”….
5. Compared to lipid nanoparticles (many research articles), what is the advantages of the cationic polymeric chitosan-based NPs?
Author Response
Reviewer 2 (Please consult the attached pdf for responses with figures Thank you)
Cystatin C is a type II cystatin and acts as an endogenous regulator of proteolytic activity by competitively and reversibly binding to the active site of several cathepsins, preventing substrate binding. In this work, the Cs-NPs were tested in the context of Mtb infection by the reference strain H37Rv and by three clinical strains with different phenotypes of drug resistance. The results show an effective capacity to silence CstC for at least 72 h during infection and following six days from the single-dose administration of Cs-NPs. This treatment resulted in improved intracellular killing for all strains, despite their different drug-resistance profiles and growth kinetics. This result is in line with our previous evidence using common commercial transfection reagents.
In summary, the results demonstrate the potential for a chitosan-nanoparticle-based solution to modulate cystatin C expression, overcome Mtb-induced blockade of the proteolytic function of macrophages and improve the control of the infection. We propose this delivery system for silencing cystatin C as a potential host-directed therapy that can be applied to complement the current antibiotic therapy and overall contribute to overcoming drug resistance. I think this paper is well organized, and the results are interesting and suitable for publication in Antibiotics. Before further consideration, some minor questions should be fully addressed.
- There are two ratios (15:1 and 2.5:1 (m/m) for GlutCs/pARG:siRNA) were employed in this work for polymeric NPs. Why did authors select these two ratios, will the ratio above 15:1 or below 2.5:1 significantly affect the resultant properties?
The ratios selected for the preparation of the polymeric NP used in this work were based on a study performed within the framework of a different project, focused on assessing the ability of GluCs or pARG to retard the mobility of a model siRNA by electrophoresis on an electrophoretic mobility shift assay (EMSA). Accordingly, different mass ratios were tested for both pARG (1:1, 1.5:1, 2:1, 2.5:1, 3:1, 3.5:1, 4:1, 4.5:1, and 5:1 (m/m)) and GluCh (1:1, 2.5:1, 5:1, 7.5:1, 10:1, 15:1, and 20:1 (m/m)). These assays indicated that the ratios 2.5:1 (m/m) and 15:1 (m/m) for pARG:siRNA and GluCs:siRNA polyplexes, respectively, enabled an adequate complexation of the siRNA, as it is shown in the images below (Matos et al, under revision).
Schematic representation of pARG:siRNA (left) and GluCs:siRNA (right) polyplex establishment, post electrostatic interaction between pARG or GluCs and siRNA, and its electrophoretic mobility at different pARG/GluCs:siRNA mass ratios, in RNA-free water. Free siRNA used as control.
- I check the Reference 12 (published by the same research group), the targeting of CstC by siRNA improves the macrophages’ response to Mtb infection. So, what’s the innovation of this work based ob polymeric NPs? In the discussion, more descriptions should be highlighted.
The authors understand the reviewer’s concern. The previous study was important to show that the targeting of the CstC by siRNA improves the macrophages response to Mtb infection. However, looking at the translation of this approach into a therapeutic strategy, alternative options should be devised to protect the oligonucleotide from degradation after administration. Accordingly, we took advantage of nanoparticles to protect the siRNA until its delivery to the target cells. The mannosylated nanoparticle that we developed was selected as the mannose moieties at its surface were shown to improve the recognition and internalization of these carriers by APC, at least in part due to the mannose receptor on the surface of these cells (Conniot et al. REF 41). Moreover, this study indicated as well that the particles followed the endolysosomal pathway, being also able to overcome endosome degradation, and thereby release payloads into the cytoplasm. As a result, this mannosylated nanoparticle would allow the protection of the siRNA upon administration, while also improving the internalization by macrophages and subsequent control of CstC gene expression.
We believe that this explanation was already present in the discussion between lines 403-409. However, the sentence below was now added (lines 409-413) to better illustrate this description.
“Our previous study indicated the ability of these mannosylated particles in following the endolysosomal pathway, in addition to being able to overcome endosomal degradation and thereby release payloads into the cytoplasm (41). Therefore, these particles were the starting point of this study focused on the regulation of CstC function and its impact on macrophages response to Mtb infection.”
We also introduced the phrase to enhance the novelty of the DS relative to common transfection methods in lines 427-445: In fact, taking into consideration that (1) common in vitro cell transfection methods are not translatable for the clinic; (2) that the present DS was demonstrated to be more efficient in silencing cystatin C mRNA along 96 h and finally (3) that it targets the siRNA to macrophages via mannose receptor were it overcomes endolysosomal degradation it is evident they have the potential for application in complex host organisms.
- Please check the Section IV (Experimental section), where did the macrophages and monocytes purchase?
As mentioned in section 4.2, the Portuguese national blood institute provided the blood used to isolate monocytes and differentiate them into macrophages. The THP-1 cell line was obtained from ATCC as indicated by the ATCC reference code. For clarity, we added the ATCC information next to the cell line.
- Some minor grammar issue. Page 8, line 270. Add a “.” in “….drug resistance The results…”….
We have corrected the sentence as the reviewer suggested.
- Compared to lipid nanoparticles (many research articles), what is the advantages of the cationic polymeric chitosan-based NPs?
We agree with the reviewer regarding the fact that several studies reported the use of lipid nanoparticles for siRNA delivery. However, also several studies report the use of polymers for efficient siRNA delivery, as an attempt to improve the stability of complexes and potentially achieve more adequate pharmacokinetics. The mannosylated particle used in this study was selected based on a previous work that showed an extensive internalization of these carriers by antigen-presenting cells, due to a combination of factors, namely the use of TPGS for particle stabilization and the presence of mannose on the surface [41] “Conniot, J.; Scomparin, A.; Peres, C.; Yeini, E.; Pozzi, S.; Matos, A.I.; Kleiner, R.; Moura, L.I.F.; Zupančič, E.; Viana, A.S.; et al. Immunization with Mannosylated Nanovaccines and Inhibition of the Immune-Suppressing Microenvironment Sensitizes Melanoma to Immune Checkpoint Modulators. Nat Nanotechnol 2019, 14, 891–901, doi:10.1038/S41565-019-0512-0). Actually, the data shown in that study points out the ability of this particle in modulating the delivery of its payloads intracellularly as it allowed to improve the presentation of antigens that follow the lysosomal pathway but also enabled the release of antigens into the cytoplasm, thus indicating endosomal escape.

Reviewer 3 Report
This manuscript evaluated the adjuvant capability of chitosan-based nanoparticle delivery system on the effective immunity-driven management of siRNA modulated cystatin C peptide at different human macrophages against different strain TB infections. The overall picture is of excellent quality, methods and results are easy to comprehend. The presented work can be considered as valuable addition to the field. Publication is recommended following issues that should be carefully addressed.
- Nanoparticle formulations should be optimized at different levels of independent variables regarding the ratios and percentages of selected formulation constituents to achieve the optimum desired particle diameter, surface charge, entrapment efficiency %, and loading capacity.
- Again, why were these concentration/ratios of NP formulation constituents were adopted as the study concentration, and is there any literature evidence or reference on which this choice was based?
- The authors performed dynamic light scattering (DLS) particle size analysis and measurement, Can the DLS graphs be incorporated in the supporting information? Likewise, if possible and available, incorporate zeta potential graphs obtained from the instrument in the manuscript or supplementary data.
- Exploring the possible interactions between cystatin C and formulation ingredients should be highlighted using Fourier transform infrared spectroscopy (ATR- FTIR).
- Moreover, morphological characterization of the formulated NPs could be investigated using transmission electron microscopy, if available.
Author Response
Reviewer 3 (please see attached file with figures)
This manuscript evaluated the adjuvant capability of chitosan-based nanoparticle delivery system on the effective immunity-driven management of siRNA modulated cystatin C peptide at different human macrophages against different strain TB infections. The overall picture is of excellent quality, methods and results are easy to comprehend. The presented work can be considered as valuable addition to the field. Publication is recommended following issues that should be carefully addressed.
We thank the reviewer for the supporting evaluation of our work.
- Nanoparticle formulations should be optimized at different levels of independent variables regarding the ratios and percentages of selected formulation constituents to achieve the optimum desired particle diameter, surface charge, entrapment efficiency %, and loading capacity.
- Again, why were these concentration/ratios of NP formulation constituents were adopted as the study concentration, and is there any literature evidence or reference on which this choice was based?
We understand the reviewer’s concerns regarding the need for a deep optimization of particle formulation regarding the ratios and percentages of selected formulation constituents to achieve the optimum desired particle diameter, surface charge, entrapment efficiency %, and loading capacity. In order to clarify this point, we would like to mention that the formulation of the PLGA-man/PLA-based particles used in this study to deliver siRNA targeting CstC was deeply optimized (volumes of the aqueous and oil phases, ratios and percentages of selected formulation constituents, emulsification time and amplitude, evaporation and centrifugation time and force) to obtain particles with near-neutral surface charges, average hydrodynamic diameters around 200 nm, low polydispersity index, and high loading capacity; in a previous study published by us ([41] “Conniot, J.; Scomparin, A.; Peres, C.; Yeini, E.; Pozzi, S.; Matos, A.I.; Kleiner, R.; Moura, L.I.F.; Zupančič, E.; Viana, A.S.; et al. Immunization with Mannosylated Nanovaccines and Inhibition of the Immune-Suppressing Microenvironment Sensitizes Melanoma to Immune Checkpoint Modulators. Nat Nanotechnol 2019, 14, 891–901, doi:10.1038/S41565-019-0512-0.”).
- The authors performed dynamic light scattering (DLS) particle size analysis and measurement, Can the DLS graphs be incorporated in the supporting information? Likewise, if possible and available, incorporate zeta potential graphs obtained from the instrument in the manuscript or supplementary data.
- Moreover, morphological characterization of the formulated NPs could be investigated using transmission electron microscopy, if available.
We appreciate the suggestion of the reviewer. Accordingly, we decided to introduce Supplementary Figure 1 containing the physicochemical (particle size) and morphological (atomic force microscopy (AFM)) characterization of the particles. Unfortunately, zeta potential values cannot be extracted from .dts file to perform the zeta potential graph.
Supplementary Figure 1. Physicochemical and morphological properties of Cs-DS. Dynamic light scattering analysis (a) and atomic force microscopy images (topography – left; phase – right) (b) showed a uniform-size polydispersity for slightly rough spherical particles.
- Exploring the possible interactions between cystatin C and formulation ingredients should be highlighted using Fourier transform infrared spectroscopy (ATR- FTIR).
We thank the reviewer for the suggestion. As it is mentioned in the answer to Reviewer 3’s first question, the formulation composition and preparation parameters have been extensively addressed in previous studies, in which oligonucleotides were also delivered, in that case, as ligands for toll-like receptor ligands on dendritic cells ([41] “Conniot, J et al. Immunization with Mannosylated Nanovaccines and Inhibition of the Immune-Suppressing Microenvironment Sensitizes Melanoma to Immune Checkpoint Modulators. Nat Nanotechnol 2019, 14, 891–901, doi:10.1038/S41565-019-0512-0”), in addition to the use of FTIR to characterize the interaction of peptides with chitosan-containing particle components (Zupancic et al. Journal of Controlled Release. 2017; 28;258:182-195. doi: 10.1016/j.jconrel.2017.05.014).

Reviewer 4 Report
To authors;
First of all, I congratulate you for your study that will contribute to the readers scientifically. The in vivo utility of therapeutics to be used in the treatment of both active TB regardless of susceptible and resistant strains and LTBI should be also evaluated. I have some suggestion on the text. It would be good to emphasize in your study that detailed studies are also needed on in vivo toxicity and excretion from the body. The article can be published after minor revision.
Thanks.
Best regards

Author Response
We thank the reviewer for the comments and corrections to the manuscript. We have altered the text to include all suggestions made by the reviewer in the pdf file and have included a commentary in the discussion about the need for further in vivo experiments.
Round 2
Reviewer 3 Report
The authors responded to inquiries. The manuscript at its current form can be considered for publication.